# Elucidating the molecular programming of a nonlinear non-ribosomal peptide synthetase responsible for fungal siderophore biosynthesis

Matthew Jenner [1,2,9] ✉, Yang Hai [3,6,9] ✉, Hong H. Nguyen[4,7], Munro Passmore[1], Will Skyrud[5,8], Junyong Kim[4], Neil K. Garg [4], Wenjun Zhang [5], Rachel R. Ogorzalek Loo [4] & Yi Tang [3]

Siderophores belonging to the ferrichrome family are essential for the viability of fungal species and play a key role for virulence of numerous pathogenic fungi. Despite their biological significance, our understanding of how these iron-chelating cyclic hexapeptides are assembled by non-ribosomal peptide synthetase (NRPS) enzymes remains poorly understood, primarily due to the nonlinearity exhibited by the domain architecture. Herein, we report the biochemical characterization of the SidC NRPS, responsible for construction of the intracellular siderophore ferricrocin. In vitro reconstitution of purified SidC reveals its ability to produce ferricrocin and its structural variant, ferrichrome. Application of intact protein mass spectrometry uncovers several non-canonical events during peptidyl siderophore biosynthesis, including inter-modular loading of amino acid substrates and an adenylation domain capable of poly-amide bond formation. This work expands the scope of NRPS programming, allows biosynthetic assignment of ferrichrome NRPSs, and sets the stage for reprogramming towards novel hydroxamate scaffolds.

Iron is an indispensable cofactor for all microbial life. The ability to coordinate and activate molecular oxygen, in addition to optimal redox properties for electron transport, places it central to numerous cellular processes[1,2]. Equally, high intra-cellular iron concentrations give rise to Fenton and Haber–Weiss reactions, producing reactive oxygen species capable of cell damage[3]. It is therefore vital that iron homoeostasis is carefully managed. Although iron has a high natural abundance, it exists predominantly as $Fe^{3+}$ in aerobic environments and tends to form insoluble ferric hydroxides rendering it inaccessible

to microorganisms[4]. As a result, organisms have evolved complex strategies for iron acquisition and storage. Whilst several mechanisms are known, a common approach employed by bacteria and fungi is the production of low-molecular-weight compounds known as siderophores, which serve as high-affinity iron chelators[5,6].

In fungi, the majority of siderophore compounds produced belong to the hydroxamate class. This functionality originates from L-ornithine, which is $N^\delta$-hydroxylated and subsequently $N^\delta$-acylated to yield either $N^\delta$-acetyl-$N^\delta$-hydroxy-L-ornithine (L-AHO) or

[1]Department of Chemistry, University of Warwick, Coventry CV4 7AL, UK. [2]Warwick Integrative Synthetic Biology Centre (WISB), University of Warwick, Coventry CV4 7AL, UK. [3]Department of Chemical and Biomolecular Engineering, University of California, Los Angeles, USA. [4]Department of Chemistry and Biochemistry, University of California, Los Angeles, USA. [5]Department of Chemical and Biomolecular Engineering, University of California, Berkeley, USA. [6]Present address: Department of Chemistry and Biochemistry, University of California, Santa Barbara, USA. [7]Present address: Transmed Co., Ltd., Ho Chi Minh City, Vietnam. [8]Present address: Arzeda, 3421 Thorndyke Ave W, Seattle WA 98119, USA. [9]These authors contributed equally: Matthew Jenner, Yang Hai. ✉e-mail: m.jenner@warwick.ac.uk; hai@chem.ucsb.edu

$N^{\delta}$-anhydromevalonyl-$N^{\delta}$-hydroxy-L-ornithine (AMHO)[7]. Typically, siderophores possess three hydroxamate units, producing a hexadentate ligand which promotes formation of a polyhedral $Fe^{3+}$ complex with binding constants in the $10^{22}$–$10^{32}$ range[8]. The hydroxamate-containing units, L-AHO and cis-/trans-AMHO, are enzymatically incorporated into chemical scaffolds and define two separate families of hydroxamate siderophores. These include the depsipeptides, typified by fusarinine C (FSC) (1)[9] and coprogen[10,11], which utilise either cis- or trans-AMHO as monomeric units and are excreted primarily to capture ferric iron (Fig. 1a)[12]. In contrast, members of the ferrichrome family, such as ferricrocin (2) and ferrichrome (3), are generally considered to be intracellular and can incorporate L-AHO or cis-/trans-AMHO, in combination with other amino acids, and are principally used for iron storage, although not exclusively (Fig. 1b, Supplementary Fig. 1)[13,14]. Both extra- and intra-cellular siderophores are essential for the survival and virulence of many problematic fungal species, including the opportunistic pathogen Aspergillus fumigatus and the rice blast fungus Magnaporthe oryzae[15,16].

Whilst the physiological function of hydroxamate siderophores in fungi is well established, in some cases, the molecular details underpinning their biosynthesis remain poorly understood. Genes encoding for large non-ribosomal peptide synthetase (NRPS) enzymes are known to be responsible for the assembly of peptidyl siderophores[10,17,18]. These modular multi-domain enzymes are typically comprised of three domain types: condensation (C), adenylation (A) and thiolation (T). During the biosynthetic process, the peptidyl intermediates are covalently tethered to the T domains via a thioester linkage, afforded by a 4′-phosphopantetheine (Ppant) moiety post-translationally appended to each T domain[19]. Within a module, the A domain specifically selects and loads an amino acid starter unit (module 1 only) or extender units onto the Ppant thiol of the T domains. This allows the C domain to catalyse amide bond formation between the growing peptidyl intermediate appended to the T domain of the upstream module, and the amino acid extender unit primed on the T domain[20,21]. Once all cycles of chain elongation are complete, the nascent peptidyl chain is cleaved from the NRPS by either a thioesterase (TE) domain, or more commonly in fungal NRPSs, a $C_T$ domain, which catalyse chain-length-specific intramolecular cyclisation to release the final product[22,23].

In bacterial NRPSs, the colinear relationship between the domain organisation and the final product allows rational assignment of biosynthetic pathways and even prediction of products from sequence data alone[21]. In contrast, fungal NRPSs typically exhibit highly aberrant domain organisations, making understanding their biosynthetic pathways challenging[18,24]. A recurring non-canonical feature of fungal NRPSs is modules lacking a functional A domain, implying that the T domain must be loaded by an A domain from a different module. This can be observed in the SidD and Nps6 NRPSs, where module 2 (M2) possesses a truncated A domain (dA) that lacks ~290 amino acids from the N-terminus rendering it catalytically inactive (Fig. 1a)[25]. Our previous work has demonstrated that the SidD $A_1$ domain loads cis-AMHO onto both the $T_1$ and $T_2$ domains, thus circumventing the requirement for an A domain in module 2[26]. Another distinctive trait is that the length and sequence of the peptide product does not correlate to the number/order of domains in the NRPS, indicative of nonlinear behaviour. This is also exemplified by the SidD NRPS, which incorporates three cis-AMHO units yet harbours only 2 modules. Our characterisation of SidD revealed that the NRPS acts in an iterative manner, loading another cis-AMHO unit onto the $T_1$ domain after formation of the di-cis-AMHO species on $T_2$, allowing a third cis-AMHO residue to be condensed and thereby yielding the fusarinine C (1) product (Fig. 1a)[26]. Whilst currently limited to the SidD NRPS, these examples of nonlinear behaviour begin to highlight the intricate and highly unusual molecular programming of fungal NRPSs.

The biosynthesis of siderophores belonging to the ferrichrome family have been linked to an evolutionarily-related group of NRPSs which exhibit unusual nonlinear domain organisations. An example of one such systems is the SidC NRPS from Aspergillus nidulans, which is known to produce ferricrocin (2) (Fig. 1b)[27]. Variations in the domain architecture of the NRPS exist, some of which give rise to the structurally related ferrichrome (3), such as Sib1 from Schizosaccharomyces pombe and Sid2 from Ustilago maydis (Supplementary Fig. 1a)[18,28,29]. The majority of ferrichrome family siderophores are cyclic hexapeptides with the exceptions of the cyclic heptapeptide, tetraglycylferrichrome, and cyclic octapeptide, epichloenin A (Supplementary Fig. 1b and c)[30,31]. Structurally, they are comprised of three $N^{\delta}$-acetyl-$N^{\delta}$-hydroxy-L-ornithine (AHO) residues for $Fe^{3+}$ chelation, and three amino acids forming a variable backbone, of which two residues are either alanine, serine or glycine, and the third a glycine residue[7]. Similar to SidD, the domain organisation of the SidC NRPS suggests a high degree of nonlinear behaviour. The SidC NRPS possesses only three A domains, yet the siderophore product requires incorporation of six amino acids. Using a linear biosynthetic logic, modules 4 and 5 both lack integrated A domains required for priming the $T_4$ and $T_5$ domains, presumably with AHO units. However, based on

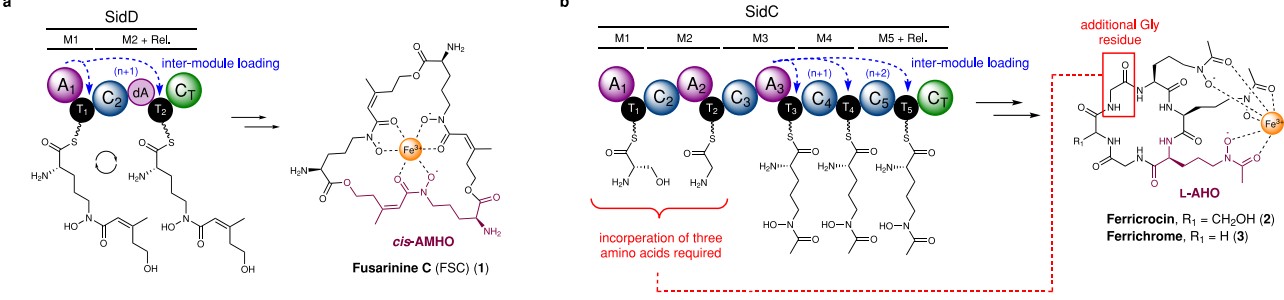

**Fig. 1 | Hydroxamate-containing siderophores produced by fungi and the nonlinear NRPSs responsible for their biosynthesis. a** Domain organisation of the SidD NRPS responsible for the biosynthesis of fusarinine C (1). The $A_1$ domain loads cis-AMHO units onto the $T_1$ and $T_2$ domains (highlighted by blue dashed arrows), a requirement due to an inactive A domain (dA) present in module 2. The NRPS acts in an iterative manner to condense three cis-AMHO units (highlighted in purple) as a depsipeptide, yielding (1) as the final product. **b** Domain organisation of the SidC NRPS responsible for the biosynthesis of ferricrocin (2). The structural variant, ferrichrome (3), is also highlighted. It is hypothesised that the $A_3$ domain loads L-AHO units (highlighted in purple) onto the $T_4$ and $T_5$ domains in a similar manner to SidD (highlighted by blue dashed arrows), as their respective modules lack dedicated A domains. The domains encompassing modules 1 and 2 must incorporate three amino acids [Gly-Ser-Gly] for (2) or [Gly-Gly-Gly] for (3) (highlighted in red). However, only two A domains are present, indicating unusual nonlinear behaviour of the NRPS. In each case, siderophores are shown in their ferric-bound state, and the hydroxymate-containing monomer unit is highlighted in purple. Domain abbreviations are as follows: C, condensation domain (dark blue); A, adenylation domain (purple); $C_T$, terminating condensation domain (green); T, thiolation domain (black).

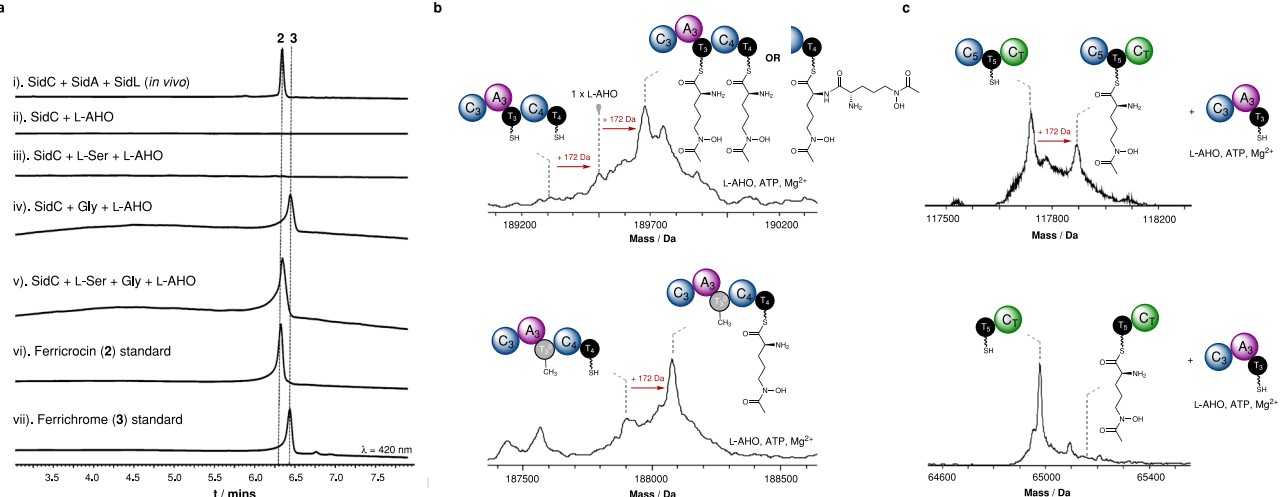

**Fig. 2 | Reconstitution of SidC NRPS and inter-modular loading of L-AHO residues by the $A_3$ domain. a** HPLC traces monitored at 420 nm for the following: (i). production of **2** via heterologous expression of *sidC*, *sidA* and *sidL* in *S. cerevisiae* JHY686; (ii)–(v). in vitro enzymatic reactions of SidC in the presence of L-AHO, +/− L-Ser and +/− Gly; (vi)–(vii). authentic standards of **2** and **3**. Presented with either a cellular pool of amino acids (i.e. in vivo experiment), or L-AHO + L-Ser + Gly in vitro, SidC produces **2** exclusively. However, when provided with L-AHO + Gly in vitro, SidC produces solely **3**. Experiments were performed in triplicate and representative spectra are shown. **b** Deconvoluted intact protein mass spectra of *holo*-SidC $C_3A_3T_3C_4T_4$ (*top*) following incubation with L-AHO, ATP and $Mg^{2+}$, showing loading of either: x2 L-AHO units onto the $T_3$ and $T_4$ domains, or a condensed di-L-AHO species on the $T_4$ domain. *holo*-SidC $C_3A_3T_3{}^0C_4T_4$ (*bottom*) following incubation

with L-AHO, ATP and $Mg^{2+}$, showing loading of a single L-AHO unit onto the $T_4$ domain. The S3151A mutation in the $T_3$ domain means it is unable to be modified with a Ppant moiety, thus preventing loading of L-AHO. **c)**. Deconvoluted intact protein mass spectra of *holo*-SidC $C_5T_5C_T$ (*top*) and *holo*-SidC $T_5C_T$ (*bottom*) following incubation with *holo*-SidC $C_3A_3T_3$, L-AHO, ATP and $Mg^{2+}$. Loading of L-AHO is only observed when the N-terminal C domain of each construct is present. Mass shifts corresponding to biosynthetic steps are highlighted with red arrows, and proposed intermediates are displayed. Markers for low abundance species are based on calculated masses or previously measured spectra. Exact measured and observed masses are detailed in Supplementary Table 2. Experiments were performed in duplicate and representative spectra are shown.

our previous observations of the SidD $A_1$ domain, we hypothesise that the SidC $A_3$ domain is likely to load the $T_4$ and $T_5$ domains with AHO units in an inter-modular manner (Fig. 1b). Assuming the $T_4$ and $T_5$ domains are indeed primed by the $A_3$ domain, under a linear paradigm this would only permit synthesis of a five-residue peptide product, suggesting further aberrant behaviour to allow incorporation of an additional Gly residue into the backbone of both ferricrocin (**2**) and ferrichrome (**3**) (Fig. 1b). It is worth noting that this level of nonlinearity has not been observed before and the exact molecular mechanism cannot be explained by any previous model.

To understand the enigmatic programming rules of this particular class of siderophore-producing NRPSs, we employ a combination of in vitro biochemical assays and intact protein mass spectrometry (MS) to interrogate SidC with respect to its ability to produce peptidyl siderophore products. Our results uncover several non-canonical events during peptidyl siderophore biosynthesis, which add previously unobserved capabilities to NRPS assembly-line enzymology and sets the stage for efforts towards reprogramming the SidC NRPS towards novel chemical scaffolds.

## Results
### Reconstitution of the SidC NRPS and determination of adenylation domain specificity

In the first instance, we elected to examine SidC activity in vivo using *Saccharomyces cerevisiae* as a heterologous host. This was conducted to allow production of the associated siderophore(s) and to ascertain whether *S. cerevisiae* would be an appropriate host for recombinant overproduction of SidC for subsequent purification and in vitro analysis. To achieve this, the *sidC* NRPS gene from *A. nidulans* FSGC A1145 (Supplementary Data 1), in addition to *sidA* and *sidL* (required for production of the L-AHO precursor), were cloned into vectors with distinct selection markers, and transformed into *S. cerevisiae* BJ5464-*npgA* (a strain with the fungal Ppant transferase, NpgA, integrated into its chromosome to ensure phosphopantetheinylation of the resulting

proteins) for siderophore production[32]. Analysis of the small molecule extract from a 3-day culture indicated that ferricrocin (**2**) was produced (Fig. 2a, trace i), and large scale cultures allowed purification and isolation of ferricrocin (**2**) for structural elucidation, which was in agreement with previous reports (see Supplementary Fig. 2–4). Having established that an active form of SidC can be produced in *S. cerevisiae*, recombinant SidC protein was overproduced in *S. cerevisiae* JHY686 as a polyhistidine-tagged fusion protein, and was purified to near-homogeneity using immobilised metal-ion affinity chromatography (IMAC) (Supplementary Fig. 5), thereby allowing controlled exposure to substrates/cofactors[33]. To ensure protein samples were completely in the *holo*-form prior to assays, purified SidC was enzymatically phosphopantetheinylated using the fungal phosphopantetheinyl transferase, NpgA (*A. nidulans*), as described previously[34]. Following addition of ATP and $Mg^{2+}$ cofactors to recombinant SidC, incubation with L-AHO alone (synthesised according to literature protocols[35,36], Supplementary Fig. 6–8), or a combination of L-AHO + L-Ser, yielded no detectable products (Fig. 2a, trace ii and iii). However, the combination of L-AHO + Gly + L-Ser resulted in the production of ferricrocin (**2**) (Fig. 2a, trace v). Interestingly, incubation with L-AHO + Gly resulted in formation of a species consistent with ferrichrome (**3**) (Fig. 2a, trace iv), which was confirmed by comparison to a chemical standard (Fig. 2a, trace vii). These observations suggest that the SidC NRPS is capable of producing both ferricrocin (**2**) and ferrichrome (**3**) depending upon the availability of amino acid substrates, yet appears to produce exclusively ferricrocin (**2**) in the native host, probably due to the abundance of L-Ser.

Our initial biosynthetic model hypothesised that each of the three A domains within SidC are responsible for activation and loading of L-Ser, Gly and L-AHO (Fig. 1b). In bacteria, bioinformatic analysis allows accurate prediction of the substrate specificity for A domains, primarily based on highly conserved amino acid motifs within the enzyme active site. However, this approach is not possible for fungal A domains, largely due to limited sequence and structural information.

Unable to predict the substrate specificity of the SidC $A_1$, $A_2$ and $A_3$ domains using bioinformatics, excised constructs of SidC $A_1$, $A_2$ and $A_3$ were cloned, overproduced in *E. coli* and purified to homogeneity for determination of substrate specificity. Using an ATP/PPi exchange assay to measure the reverse reaction of the adenylation step, the SidC $A_1$ domain showed activation of L-Ala and L-Ser, with a clear preference towards the latter (Supplementary Fig. 9, top). Interestingly, the SidC $A_2$ domain exhibited no detectable activity when subjected to the ATP/PPi exchange assay, suggesting that PPi may not formed during the activation step. However, activity was observed for SidC $A_2$ using the hydroxylamine release assay, which indicates formation of the corresponding aminoacyl-adenylate, and allowed a substrate specificity profile to be obtained revealing Gly as the preferred substrate (Supplementary Fig. 9, middle). The discrepancy between these two assays indicates that the $A_2$-catalysed adenylation reaction is not reversible[37,38]. Activity for the SidC $A_3$ domain was obtained using the ATP/PPi exchange assay and displayed a clear preference towards L-AHO (Supplementary Fig. 9, bottom). The SidC $A_1$ and $A_2$ assays were also reconstituted with their cognate $T_1$ and $T_2$ domains to examine the aminoacyl transfer step. Intact protein mass spectrometry (MS) analysis showed mass shifts corresponding to loading of L-Ser and Gly onto the respective T domains (Supplementary Fig. 10).

### The SidC $A_3$ domain catalyses *intra*- and *inter*-modular loading of L-AHO

The biosynthesis of ferricrocin (**2**) and ferrichrome (**3**) both require installation of three L-AHO residues, yet the SidC NRPS possesses only a single A domain capable of activating L-AHO; the $A_3$ domain situated in module 3. Furthermore, modules 4 and 5 lack integrated A domains for loading amino acid units to their cognate T domains (Fig. 1b). Based on our previous observations in the SidD NRPS, we postulated that the SidC $A_3$ domain may be capable of loading L-AHO units onto the $T_4$ and $T_5$ domains, in addition to its cognate $T_3$ domain. This would result in the $T_3$, $T_4$ and $T_5$ domains being charged with L-AHO units, which can then be condensed together by the sequential activity of the $C_4$ and $C_5$ domains to yield the tri- L-AHO motif. An alternative model would involve iterative activity of module 3 to generate tri-L-AHO appended to the $T_3$ domain; however, this would render the $C_4T_4C_5T_5$ region redundant for the biosynthesis and seemed less likely.

To investigate this aspect, a SidC $C_3A_3T_3$ tri-domain construct was cloned, overproduced and purified to examine the covalently tethered intermediates loaded onto the $T_3$ domain. Following conversion to its *holo* form and subsequent incubation with ATP, $Mg^{2+}$ and L-AHO (15 min), intact protein MS analysis of $C_3A_3T_3$ revealed loading of a single L-AHO unit, indicated by a + 172 Da mass shift relative to the mass of *holo*-$C_3A_3T_3$ (Supplementary Fig. 11). This highlighted that the standalone SidC $C_3A_3T_3$ tri-domain is only capable of loading a single L-AHO unit onto its cognate $T_3$ domain and ruled out the possibility of iterative loading. We next generated a SidC $C_3A_3T_3C_4T_4$ penta-domain construct to examine the ability of the $A_3$ domain to catalyse loading of L-AHO onto the $T_4$ domain. Under the same conditions, two sequential +172 Da mass shifts were observed in the mass spectrum, congruent with loading of two L-AHO units (Fig. 2b, top). The measured masses were consistent with either two L-AHO units loaded in an uncondensed form onto the $T_3$ and $T_4$ domains, or in a condensed di-L-AHO species on the $T_4$ domain. To validate these observations, a mutant of the SidC $C_3A_3T_3C_4T_4$ construct was produced, where the Ser residue that serves as the Ppant group attachment site was mutated to Ala (S3140A, designated as $T_3{}^0$), allowing only the $T_4$ domain to be converted to its *holo* form. When subjected to the loading assay, the SidC $C_3A_3T_3{}^0C_4T_4$ protein was able to activate and transfer an L-AHO unit onto the $T_4$ domain, indicated by a single +172 Da mass shift in the intact protein mass spectrum (Fig. 2b, bottom).

Inter-modular loading of $T_4$ by the $A_3$ domain in the SidC NRPS is reminiscent of behaviour observed for the SidD NRPS, where the $A_1$

domain is able to prime $T_2$, situated in the downstream module. In both of these instances, the A domain is interacting with a T domain situated one module downstream. However, in the SidC NRPS, we hypothesised that the $A_3$ domain is capable of loading the $T_5$ domain, situated two modules downstream. To probe this, we conducted a bimolecular assay between SidC $C_3A_3T_3$ and SidC $C_5T_5C_T$ in the presence of ATP, $Mg^{2+}$ and L-AHO. This resulted in ~35% of L-AHO loading onto the $T_5$ domain after a 60 min incubation (Fig. 2c, top), and indicated that the $A_3$ domain is capable of loading the $T_5$ domain, situated two modules downstream. An equivalent experiment using a $C_4T_4$ construct yielded comparable levels of L-AHO loading, serving as a control measurement, and also highlighting the reduced efficiency when domains are not covalently tethered in megasynth(et)ases (Supplementary Fig. 12). Interestingly, spectra obtained from this experiment also gave rise to a small peak congruent with a di- L-AHO species. The relatively small amount of this condensed species relative to the mono- L-AHO suggests that the $C_3$ domain does not preferentially condense L-AHO units together, indicating that the species in Fig. 2b (*top*) is likely two uncondensed L-AHO units. Analogous assays using the SidC $T_5C_T$ didomain and SidC $T_4$ domain (i.e. without the N-terminal C domain), resulted in no detectable L-AHO loading (Fig. 2c, *bottom* and Supplementary Fig. 12), implying that the N-terminal C domains facilitate the loading reaction. These results suggest two architectural models for non-linear L-AHO loading by the $A_3$ domain. One possibility is that intra-chenar loading of L-AHO is promoted by a 3-dimensional arrangement of the SidC NRPS that enables proximity of the $A_3$ domain to the $T_4$ and $T_5$ domains (Supplementary Fig. 13a). Here, the presence of the $C_4$ and $C_5$ domains may be essential to provide an interaction 'platform' for the $T_4$ and $T_5$ domains to access the $A_3$ domain. A second possibility involves inter-chenar communication between two SidC proteins, allowing the $A_3$ domain to load L-AHO onto the $T_4$ and $T_5$ domain in trans, whilst loading the $T_3$ domain conventionally (Supplementary Fig. 13b). Interestingly, modelling of the $A_3T_3C_4T_4C_5T_5$ region using AlphaFold[39] suggests that both $C_4$ and $C_5$ domains form interfaces with the $A_3$ domain, conceivably providing a platform for their respective T domains to access the $A_3$ domain active site, adding credence to an intra-chenar loading model (Supplementary Figs. 13c, 14 and 15).

### SidC $C_2A_2T_2$ tri-domain catalyses non-canonical loading of Gly residues

We next turned our attention to the biosynthetic steps required for the formation of the tripeptide backbone. This region is the sole structural difference between ferricrocin (**2**) and ferrichrome (**3**), possessing [Gly]-[L-Ser]-[Gly] and [Gly]-[Gly]-[Gly] motifs, respectively. The SidC A domain specificity assays determined that the $A_1$ domain preferentially activates L-Ser, and that the $A_2$ domain only activates Gly (Supplementary Fig. 9, 10). Based on these observations, linear assembly of the amino acid units would yield a $T_2$-[Gly]-[L-Ser]-$NH_2$ species, requiring a second Gly to be non-canonically condensed onto the amine of L-Ser to yield the $T_2$-[Gly]-[L-Ser]-[Gly]-$NH_2$ tripeptide intermediate necessary for ferricrocin (**2**) production. However, for ferrichrome (**3**), two scenarios seemed plausible: i). the $A_1$ domain would instead load Gly onto the $T_1$ domain (note, some activity towards Gly observed in specificity assays (Supplementary Fig. 9)), allowing a $T_2$-[Gly]-[Gly]-$NH_2$ species to be formed, followed by non-canonical condensation of a third Gly to yield the $T_2$-[Gly]-[Gly]-[Gly]-$NH_2$ intermediate. ii). the $A_1$ and $T_1$ domains are not utilised, leaving the $A_2$ domain to generate a $T_2$-[Gly]-$NH_2$ species, which must undergo two sequential non-canonical condensation events to yield the $T_2$-[Gly]-[Gly]-[Gly]-$NH_2$ intermediate.

In order to unpick these biosynthetic steps, we first produced a SidC($\Delta A_1T_1$) construct to examine whether the domains of module 1 are essential for siderophore production. Using *S. cerevisiae* as a heterologous host, SidC ($\Delta A_1T_1$) was observed to produce ferrichrome (**3**) exclusively, with no ferricrocin (**2**) detected (Fig. 3a, trace i), which was

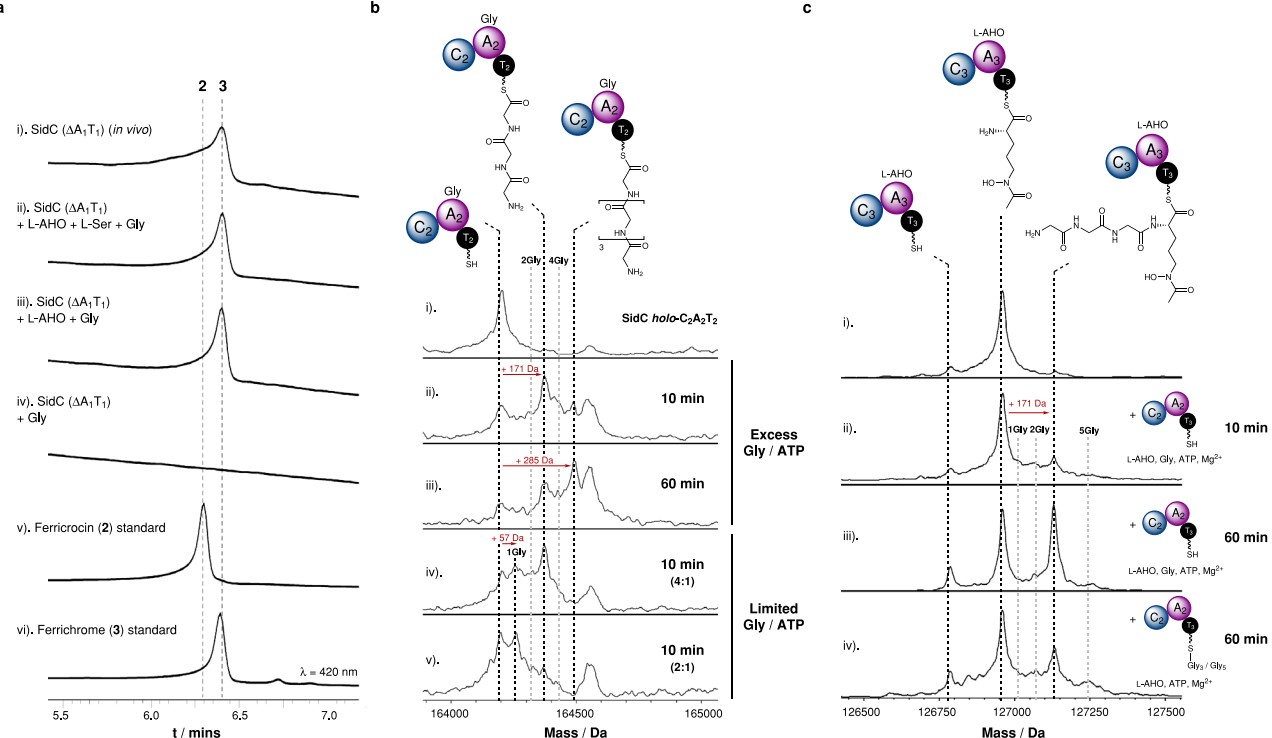

**Fig. 3 | Iterative loading of Gly residues by the $A_2$ domain and chain-length control by the $C_3$ domain. a** HPLC traces monitored at 420 nm for the following: (i). production of **3** via heterologous expression of *sidC* ($\Delta A_1 T_1$), *sidA* and *sidL* in *S. cerevisiae* JHY686; ii - iv). in vitro enzymatic reaction of SidC ($\Delta A_1 T_1$) in the presence of Gly, +/− L-Ser and +/− L-AHO; (v)–(vi). authentic standards of **2** and **3**. Experiments were performed in triplicate and representative spectra are shown. **b** Stacked deconvoluted intact protein mass spectra of *holo*-SidC $C_2A_2T_2$ in the presence of Gly, ATP and $Mg^{2+}$. Assays conducted with excess Gly/ATP are shown in spectra (ii.) and (iii.) at 10 min and 60 min time intervals, and with limited concentrations of Gly/ATP in spectra (iv.) and (v.) after a 10 min incubation. Mass shifts corresponding to mono-/poly-Gly species are highlighted with red arrows, and proposed intermediates are displayed. **c** Stacked deconvoluted intact protein mass spectra of L-AHO-SidC $C_3A_3T_3$ following incubation with *holo*-SidC $C_2A_2T_2$, Gly, ATP and $Mg^{2+}$ following 10 min and 60 min incubation periods. Spectrum (i.) shows L-AHO SidC

$C_3A_3T_3$ alone; spectra (ii.) and (iii.) show increasing production of the condensed product, L-AHO-Gly$_3$-SidC $C_3A_3T_3$ over time. Only the Gly$_3$ condensed product is observed, not Gly$_1$ or Gly$_2$, suggesting that this is not a stepwise process. Instead, Gly$_3$ must be formed on SidC $C_2A_2T_2$ before the SidC $C_3$ domain will catalyse the condensation reaction. Spectrum iv. shows an experiment where a 60 min pre-incubation of *holo*-SidC-$C_2A_2T_2$ with Gly, ATP and $Mg^{2+}$ was conducted to allow formation of Gly$_3$/Gly$_5$-SidC $C_2A_2T_2$ (see Fig. 3b, spectrum (iii.), before addition of L-AHO-SidC $C_3A_3T_3$. Only the Gly$_3$ condensed product is observed, not Gly$_5$, indicating that the SidC $C_3$ domain selectively condenses the Gly$_3$-SidC $C_2A_2T_2$ species with L-AHO only. Mass shifts corresponding to biosynthetic species are highlighted with red arrows, and proposed intermediates are displayed. Exact measured and observed masses are detailed in Supplementary Tables 2 and 3. Experiments were performed in duplicate and representative spectra are shown.

the product of full-length SidC under the same culture conditions (Fig. 2a). Purification of the recombinant SidC($\Delta A_1 T_1$) protein allowed controlled exposure to amino acid substrates. Here, SidC($\Delta A_1 T_1$) produced ferrichrome (**3**) exclusively, provided that both Gly and L-AHO were present (Fig. 3a, traces ii–iii). The inclusion of L-Ser did not promote ferricrocin (**2**) production (Fig. 3a, trace ii), and omission of L-AHO resulted in no detectable products (Fig. 3a, trace iv). These data indicated that the $A_1 T_1$ domains are essential for ferricrocin (**2**) production, but are not required for ferrichrome (**3**) formation. Furthermore, these data indicated that during ferrichrome (**3**) biosynthesis, neither $A_1$ or $T_1$ domain participate in the recruitment of the additional Gly residue. This left the intriguing possibility that module 2 alone (i.e. $C_2A_2T_2$) could be responsible for constructing the $T_2$-[Gly]-[Gly]-[Gly]-$NH_2$ intermediate required for ferrichrome (**3**) biosynthesis.

To explore this prospect, we cloned an MBP-SidC $C_2A_2T_2$ fusion construct, which was overexpressed in *E. coli* as a soluble protein and purified to homogeneity for in vitro studies. Using our established intact protein MS approach, incubation of SidC $C_2A_2T_2$ with excess Gly, ATP and $Mg^{2+}$ for 10 mins led to emergence of two new peaks which were + 171 Da and + 285 Da relative to the *holo*-$C_2A_2T_2$ species, mass shifts which correspond to condensed tri- and penta-Gly chains, respectively, attached to the $T_2$ domain (Fig. 3b, spectra i–ii). Upon extension of the incubation period to 60 min, the relative abundance

of the penta-Gly species increased as the tri-Gly decreased, suggesting that the tri-Gly intermediate is a precursor to the penta-Gly chain (Fig. 3b, spectrum iii). In the presence of excess Gly and ATP conversion to the tri-Gly and penta-Gly intermediates was fast, preventing observation of early intermediates of the poly-Gly chain. Therefore, assays were repeated with reduced relative concentrations of 4:1 and 2:1 (Gly/ATP:protein) to capture early-stage intermediates. At a 4:1 ratio, a peak at + 57 Da relative to the *holo*-$C_2A_2T_2$ species was observed, which correlates to a single Gly unit loaded onto the $T_2$ domain, in addition to the previously observed tri-Gly species (Fig. 3b, spectrum iv). Switching to a 2:1 ratio yielded predominantly the mono-Gly species, with low levels of di- and tri-Gly intermediates observable in the spectrum (Fig. 3b, spectrum v).

These observations indicated that the domains of module 2 alone are capable of constructing Ppant-bound poly-Gly chains, appearing to favour the formation of tri- and penta-Gly species. While loading of a single Gly unit onto the $T_2$ domain is likely catalysed by the $A_2$ domain in a canonical fashion, the mechanism by which the remaining two Gly units are condensed onto the free amine remained elusive. The use of an isolated $C_2A_2T_2$ construct in our experiments, meant it was possible that inter-chenar communication between individual $C_2A_2T_2$ proteins could allow the $C_2$ domain to catalyse condensation reactions between Gly chains, assuming that the $T_2$ domain is able to act as an

aminoacyl donor and acceptor (i.e. $C_2A_2T_2$-[Gly]-$NH_2$ + $C_2A_2T_2$-[Gly]-$NH_2$ → $C_2A_2T_2$-SH + $C_2A_2T_2$-[Gly]-[Gly]-$NH_2$). To test this possibility, the catalytically essential active site $His_{569}$ residue of the $C_2$ domain was mutated to Ala to produce a catalytically inactive $C_2$ domain, denoted as $C_2^0$. Upon incubation with Gly and co-factors, the SidC $C_2^0A_2T_2$ protein was able to generate poly-Gly chains in a near-identical manner to that of the wild-type construct (Supplementary Fig. 16). We attempted to produce a SidC $A_2T_2$ construct, however, in the absence of the N-terminal $C_2$ domain the resulting protein was highly unstable and degraded quickly, suggesting that the C domain plays an important structural role within the module.

Taken together, our data strongly indicate that the SidC $A_2$ domain is responsible for catalysing both canonical loading of a single Gly unit onto the Ppant thiol of the $T_2$ domain, and subsequent amide bond formation steps to generate the tri-Gly intermediate required for ferrichrome (**3**) biosynthesis. Whilst two non-canonical amide bond-forming steps are required for ferrichrome (**3**) biosynthesis, this is only required once for ferricrocin (**2**) biosynthesis, adding a Gly to the free amine of the $T_2$-[Gly]-[L-Ser]-$NH_2$ species. Poly-amide bond formation catalysed by an A domain is unprecedented within the context of a multi-modular NRPS, making this system particularly interesting. However, similar activity has been observed for a standalone A domain during the biosynthesis of streptothricin. Here, following canonical loading of a L-β-lysine residue onto a T domain, a separately encoded adenylation domain, ORF19, generates poly-L-β-lysine chains via amide bond formation with the free ε-$NH_2$ group (Supplementary Fig. 17)[40]. Interestingly, the SidC $A_2$ domain differs from ORF19, in that the amide bond is formed using the α-$NH_2$ group and the domain is found integrated into the NRPS. It is worth noting that stand-alone adenylation domains have also been observed to catalyse amide bond formation in the biosynthesis of pacidiamycin (PacU), coumermycin $A_1$ (CouL) and novobiocic acid (NovL) (Supplementary Fig. 17)[41–43]. However, these examples involve single condensation events, not formation of poly-amino acid chains as observed for the ORF19 and SidC $A_2$ domains.

### The SidC $C_3$ domain is a chain-length gatekeeper

Our biochemical investigations of SidC $C_2A_2T_2$ demonstrated its ability to generate poly-Gly chains of up five residues in length (Fig. 3b). However, the biosynthetic products of SidC, ferricrocin (**2**) and ferrichrome (**3**), both require the $T_2$-tethered intermediate to be three residues in length: $T_2$-[Gly]-[L-Ser]-[Gly]-$NH_2$ and $T_2$-[Gly]-[Gly]-[Gly]-$NH_2$, respectively. Therefore, in order to maintain biosynthetic fidelity, we postulated that the SidC $C_3$ domain imposes a selective requirement for three-residue chains appended to the $T_2$ domain in order to catalyse condensation with the first L-AHO unit, effectively acting as a gatekeeper. To explore this hypothesis, we incubated *holo*-SidC $C_2A_2T_2$ with *holo*-SidC $C_3A_3T_3$ in the presence of Gly, L-AHO, ATP and $Mg^{2+}$, and monitored the SidC $C_3A_3T_3$ protein using intact protein MS at several time points. After 10 min, a new peak at + 171 Da from the L-AHO-$C_3A_3T_3$ species had emerged, indicating condensation of a tri-Gly unit onto the L-AHO (Fig. 3c, spectra (i) and (ii)), with the intensity of this species increasing over a 60 min period (Fig. 3c, spectrum (iii)). The absence of signals corresponding to mono- (+ 57 Da) or di-Gly (+ 114 Da) species condensed with L-AHO-$C_3A_3T_3$ during the time-course strongly indicated that the entire $T_2$-[Gly]-[Gly]-[Gly] intermediate is condensed with L-AHO, rather than sequential addition of Gly residues.

To examine whether the SidC $C_3$ domain can discriminate between tri-, tetra- and penta-Gly intermediates, we pre-incubated *holo*-SidC $C_2A_2T_2$ with Gly, ATP and $Mg^{2+}$ for 60 min to generate a mixture of poly-Gly chain lengths, represented by Fig. 3c, spectrum iii. The remaining Gly in the reaction was then removed by multiple cycles of ultrafiltration, before addition to *holo*-SidC $C_3A_3T_3$ in the presence

of L-AHO, ATP and $Mg^{2+}$ for 60 min. Subsequent intact protein MS analysis revealed only the tri-Gly species condensed with $T_3$-tethered L-AHO (Fig. 3c, spectrum iv), suggesting that the SidC $C_3$ domain possesses strict selectivity for poly-amino acid chain lengths where $n = 3$, thereby acting as a critical checkpoint during the biosynthesis.

### A biosynthetic model for siderophore production by the SidC NRPS

Our data allows proposal of a rational biosynthetic model for the construction of ferricrocin (**2**) and ferrichrome (**3**) by the SidC NRPS (Fig. 4). In ferricrocin (**2**) biosynthesis, the process is initiated by the $A_1$ domain loading a L-Ser residue onto the $T_1$ domain, which is subsequently condensed with a Gly residue tethered to the downstream $T_2$ domain by the $C_2$ domain, as a result of $A_2$ domain loading, yielding the $T_2$-[Gly]-[L-Ser]-$NH_2$ intermediate (Fig. 4a). This initial step is not required for ferrichrome (**3**) biosynthesis, which commences with $A_2$ domain-catalysed loading of a single Gly residue onto the $T_2$ domain, and is then condensed with a second Gly residue, catalysed by the amide bond-forming capabilities of the $A_2$ domain, yielding a $T_2$-[Gly]-[Gly]-$NH_2$ intermediate (Fig. 4b). Both $T_2$-tethered dipeptide intermediates during ferricrocin (**2**) and ferrichrome (**3**) biosynthesis then undergo addition of a Gly residue to the free $NH_2$ group, catalysed by the $A_2$ domain, producing $T_2$-[Gly]-[L-Ser]-[Gly]-$NH_2$ and $T_2$-[Gly]-[Gly]-[Gly]-$NH_2$ intermediates. Whilst the $A_2$ domain is capable of adding further Gly residues to extend the peptidyl chain over time (Fig. 3b), the nascent tripeptide intermediates are rapidly and selectively condensed with the $T_3$-tethered L-AHO species by the $C_3$ domain.

## Discussion

Ferrichrome NRPSs are found in the vast majority of Ascomycetes, providing the biosynthetic machinery for siderophore production. Despite producing near-identical products, ferricrocin (**2**) and ferrichrome (**3**), substantial differences in the NRPS domain architecture exist within the NRPS family (Types I–V, Fig. 5). Phylogenetic work has suggested that ferrichrome NRPSs originate from an ancestral colinear hexamodule NRPS, created by adjacent duplication of complete NRPS modules resulting in two lineages: NSP2 and NSP1/SidC. The recently reported Sid1 NRPS responsible for AS2488059 biosynthesis, a related ferrichrome siderophore, might be considered as a contender for this ancestral gene (Supplementary Fig. 18)[44–46]. Here, dedicated A domains are employed to load each of the three L-AHO residues, in addition to the three backbone residues (Asn, Val and Phe), totalling six A domains in the NRPS. However, despite the similarities, phylogenetic analyses suggest that the Sid1 NRPS is of a different evolutionary origin to siderophores of the ferrichrome family[46]. All combinations of the ferrichrome family of NRPSs give rise to unusual non-linear domain organisations, which cannot be reconciled with standard biosynthetic logic of NRPSs. Plausible biosynthetic proposals linking the domain organisation to the peptidyl product require inter-modular loading of amino acid substrates by A domains up to $n + 2$ modules downstream (blue arrows), and/or A domains capable of creating (poly)amide chains on the same T domain (red arrows). Our study of the SidC NRPS highlights that both activities are possible in NRPSs and allow evidence-based biosynthetic proposals for all variations of the ferrichrome NRPS (Fig. 5).

Members of the NPS2 lineage (Types III, IV and V) all possess the correct number of C domains required for the number of amide bonds formed in the peptidyl product. However, loss or degeneration of A domains requires inter-module loading of T domains with amino acid substrates. This is observed in the tri-Orn region, as characterised for SidC, and in the tripeptide region for Type III and IV. In contrast, the Type I and II members of the NPS1/SidC lineage both require an amide bond-forming A domain to compensate for the lack of C domains in the NRPS, in addition to inter-module loading capabilities in the tri-Orn region. The CsNPS2 and Cpf1 NRPSs (Type VI) are the most truncated

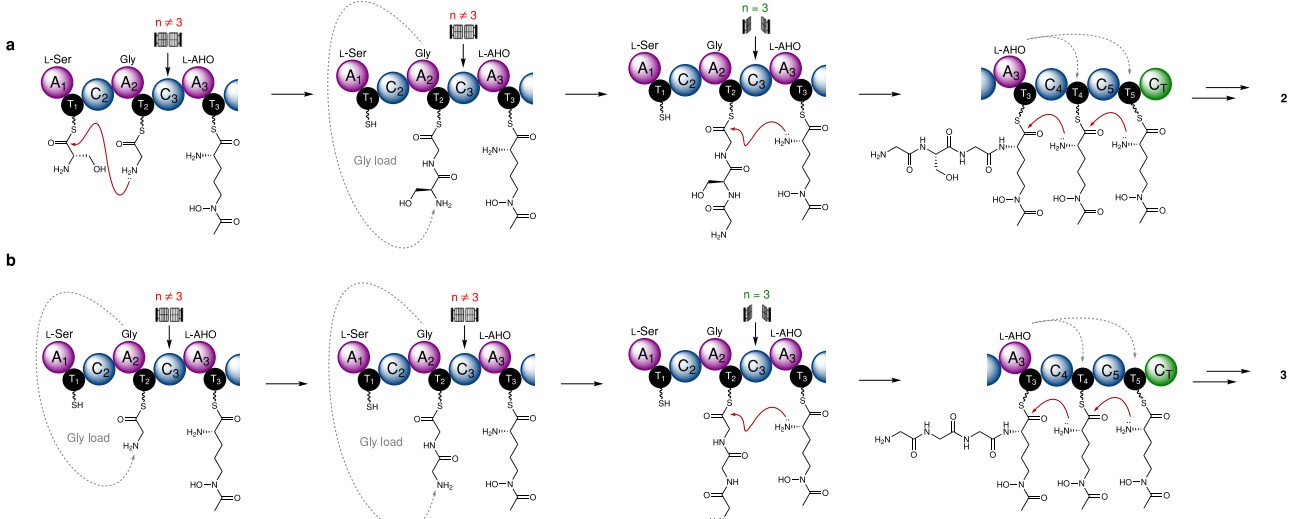

**Fig. 4 | Proposed biosynthetic models for SidC-catalysed formation of ferri-crocin and ferrichrome. a** Ferricrocin (**2**) biosynthesis commences with condensation between L-Ser and Gly, which is catalysed by the SidC $C_2$ domain forming an (L-Ser)-Gly dipeptide ($n = 2$) tethered to the SidC $T_2$ domain. Non-canonical ligation of a Gly unit onto the amine of L-Ser produces a Gly-(L-Ser)-Gly tripeptide ($n = 3$), which can undergo condensation with L-AHO catalysed by the chain-length selective SidC $C_3$ domain. The SidC $A_3$ domain loads L-AHO onto SidC $T_4$ and $T_5$ domains allowing a succession of condensation events to generate a Gly-(L-Ser)-Gly-

(L-AHO)$_3$ hexapeptide intermediate bound to the SidC $T_5$ domain. Chain release is catalysed by the C-terminal $C_T$ domain to yield the biosynthetic product **9**.
**b** Ferrichrome (**3**) biosynthesis can occur in the absence of L-Ser, where canonical loading of Gly onto the $T_2$ domain is followed by two successive rounds of non-canonical Gly ligation to yield a Gly$_3$ species ($n = 3$) tethered to the $T_2$ domain. The remaining steps are identical to the biosynthesis of **2**, to yield the biosynthetic product **3**.

variation and appears to have lost much of the N-terminus, leaving just the tri-Orn region that requires inter-module loading capabilities. Recent assignments of the CsNPS2 and Cpf1 products as basidioferrin (**4**) and coprinoferrin (**5**) revealed a structure comprised of three condensed L-AHO units for basidioferrin (**4**)[47], and three L-hydroxyhexanoyl ornithine (L-hhOrn) units for coprinoferrin (**5**)[48]. In the latter, this suggests that the Cpf1 $A_1$ domain (equivalent to $A_3$ in SidC and Sid2) has evolved specificity towards a larger hydroxamate substrate, whilst retaining the ability to conduct inter-module loading (Fig. 5).

Taken together, our observations highlight the impressive evolutionary changes employed by fungal NRPSs to improve atom economy and increase structural diversity in their biosynthetic assembly-lines. Our improved understanding of the biosynthetic rules has set the stage for manipulating and recombining these pathways towards novel hydroxamate-containing scaffolds.

## Methods
### Molecular cloning and site directed mutagenesis
Yeast Expression Constructs: the SidC (Genbank: XM_653119 [https://www.ncbi.nlm.nih.gov/nuccore/XM_653119.2]), SidA (Genbank: XM_658335 [https://www.ncbi.nlm.nih.gov/nuccore/XM_658335.1]), and SidL (Genbank: XM_652967 [https://www.ncbi.nlm.nih.gov/nuccore/XM_652967]) gene exon fragments were cloned from the cDNA library prepared from the mRNA extract of *A. nidulans* FSGC A1145 strain[49] cultured on Czapek-Dox (CD) agar. The corresponding yeast expression plasmids were assembled through yeast homologous recombination using a Frozen-EZ Yeast Transformation II Kit (Zymo research). Gene fragments were integrated into a 2μ-based yeast expression vector with auxotrophic markers and ADH2 promoter and terminator regions. All proteins were cloned in-frame with an N-terminal pHis$_8$ tag to facilitate purification. *E. coli* Expression Constructs: target regions of SidC were subcloned into either pHis$_6$-MBP-pET28a or pHis$_6$-pET28a vectors. All proteins were cloned in-frame with an N-terminal TEV-cleavable tag (either MBP or pHis$_6$), allowing removal post-purification. Primers used for the cloning of SidC

constructs and mutagenic primers to generate point-mutations/truncations are detailed in Supplementary Table 1.

### Protein overproduction and purification
**Yeast expression constructs.** The full-length proteins were expressed in *S. cerevisiae* JHY686[33] strain cultured in YPD medium. Briefly, single colonies of yeast cells harbouring expression plasmids were inoculated into SDCt uracil drop-out culture and left growing at 28 °C for 2 days. The seed culture was then inoculated into YPD culture (20 ml to 1000 mL) and left growing at 28 °C for another 2 days. Cells were harvested by centrifugation and washed once with cell lysis buffer (50 mM K$_2$HPO$_4$ (pH 7.5), 10 mM imidazole, 300 mM NaCl, 5% glycerol). Cells were flash frozen in liquid nitrogen and lysed by using a stainless-steel Waring blender. The cell lysate was cleared by centrifugation at 26,000$g$ for 60 min at 4 °C and the supernatant was filtered through a 0.22 μm filter (Millipore). The filtrate was incubated with Ni$^{2+}$-NTA resin for 30 min at 4 °C and then the slurry was loaded onto a gravity column. The resin was washed and eluted with increasing concentrations of imidazole in cell lysis buffer. The fractions were examined by SDS-PAGE gels. Pure fractions were concentrated to ~20 mg/mL by Amicon concentrators (Millipore), supplemented with 10% glycerol and stored at −80 °C. Protein concentrations were determined by Bradford assay. Typically, 2 L cell culture could yield 1 − 10 mg of protein depending on the nature of the protein construct.

**E. coli expression constructs.** A single colony of *E. coli* BL21 (DE3) that had been transformed with the appropriate expression vector was picked and used to inoculate LB medium (5 or 10 mL) containing kanamycin (50 μg/mL). The resulting culture was incubated overnight at 37 °C and 180 rpm then used to inoculate LB medium (0.5 or 1 L) containing kanamycin (50 μg/mL). The resulting culture was incubated at 37 °C and 180 rpm until the optical density of the culture at 595 nm reached 0.6, then IPTG (1 mM) was added and growth was continued overnight at 15 °C and 180 rpm. The cells were harvested by centrifugation (4000$g$, 15 min, 4 °C) and re-suspended in buffer (20 mM Tris-HCl, 100 mM NaCl, 20 mM Imidazole, pH 7.4) at 10 mL/L of growth

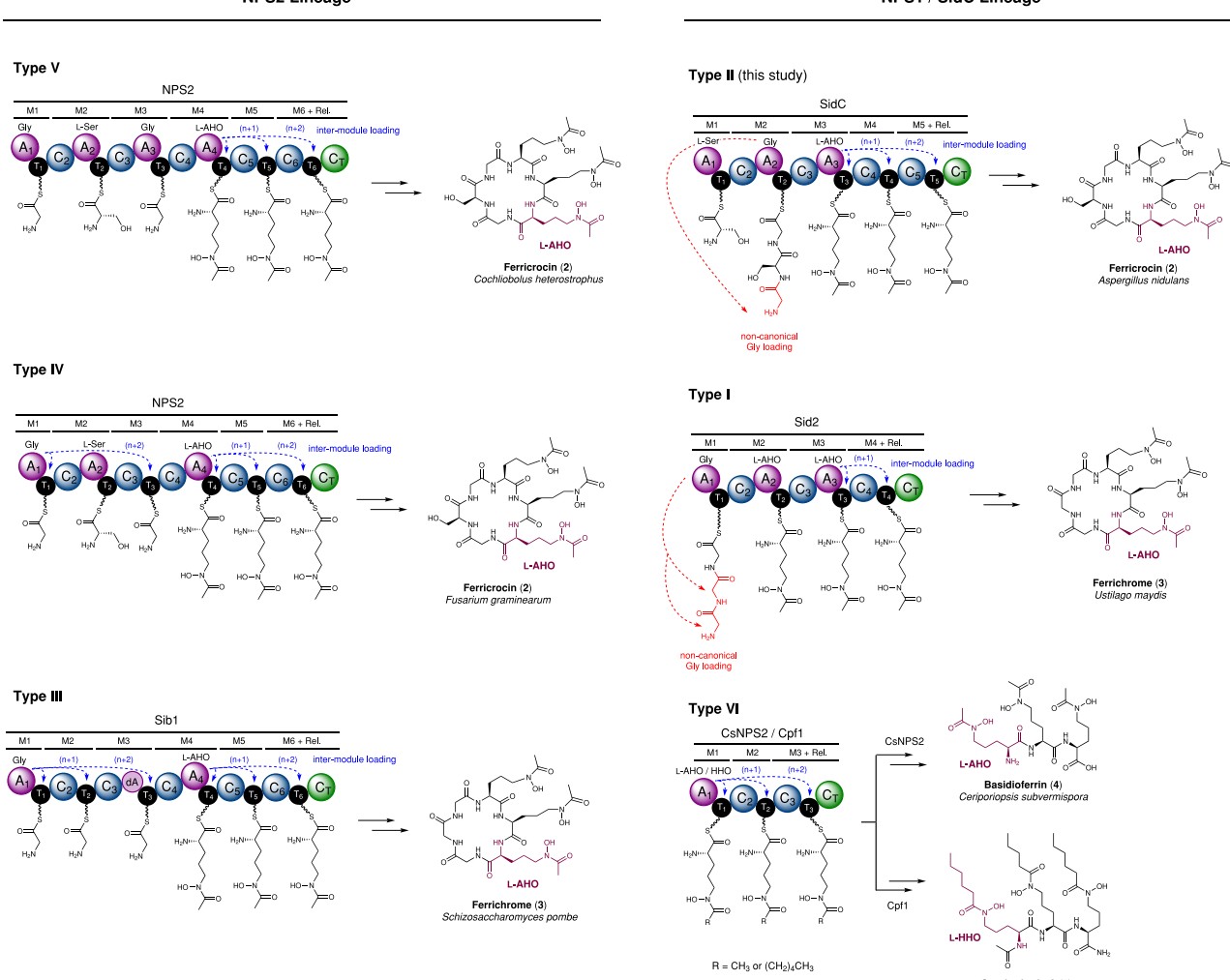

**Fig. 5 | Biosynthetic schemes for the six modular architectures of ferrichrome class of NRPSs.** The new programming rules allow the biosynthetic assignment of other architectures of ferrichrome-family synthetase NRPSs. Examples include: Sid2 (ferrichrome), *U. maydis*;[53] SidC (ferricrocin), *A. nidulans*;[27] Sib1 (ferrichrome), *S. pombe*;[17,54] NPS2 (ferricrocin), *F. graminearum*;[55] NPS2 (ferricrocin), *C. heterostrophus*;[56] CsNPS2 (basidioferrin), *C. subvermispora*;[47] and Cpf1 (coprino-ferrin A), *C. cinerea*[48]. The inter-module loading events, either (*n* + 1) or (*n* + 2), are highlighted in blue, and non-canonical loading of Gly residues is highlighted in red. In each case, siderophores are shown in their desferric state, and the hydroxymate-containing monomer unit is highlighted in purple. The lineage classification and Type I–VI groupings are based on previous phylogenetic analyses of ferrichrome synthetase NRPSs conducted by Bushley et al.[18] All biosynthetic proposals are based on observations from the SidC NRPS in this study, but other possibilities may exist.

medium then lysed using a Constant Systems cell disruptor. The lysate was centrifuged (37,000*g*, 30 min, 4 °C) and the resulting supernatant was loaded onto a HiTrap FF Chelating Column (GE Healthcare), which had been pre-loaded with 100 mM NiSO$_4$ and equilibrated in re-suspension buffer (20 mM Tris-HCl, 100 mM NaCl, 20 mM Imidazole, pH 7.4). Proteins were eluted in a stepwise manner using re-suspension buffer containing increasing concentrations of imidazole−50 mM (5 mL), 100 mM (3 mL), 200 mM (3 mL) and 300 mM (3 mL). The presence of the protein of interest in fractions was confirmed by SDS-PAGE, and an additional gel filtration step (Superdex 75/200, GE Healthcare) was used to further purify proteins where necessary. Fractions containing the protein of interest were pooled and concentrated to 250−400 μM using a Viva-Spin centrifugal concentrator (Sartorius) at an appropriate MWCO. Samples were snap-frozen in liquid N$_2$ and stored at −80 °C.

**Siderophore isolation/preparation**

Desferriferricrocin and ferricrocin were obtained through coexpression of *sidC*, *sidA*, and *sidL* genes in *S. cerevisiae* BJ5464-npgA strain[50]. Briefly, competent yeast cells were transformed with plasmids XW55-SidC, XW06-SidL and XW02-SidA and the colonies harbouring these three plasmids were selected using minimal medium dropping out uracil, tryptophan, and leucine. The colony was inoculated into the corresponding liquid minimal medium, and the cell culture was grown at 28 °C for 2 days. To induce production, the starting culture was inoculated to YPD medium and left growing at 28 °C for 3 days. The cell pellet was harvested through centrifugation and the produced siderophore compound was extracted using acetone. The organic extract was dried using rotary evaporation and the residue was dissolved in methanol and subjected to LC-MS analysis on a Shimadzu 2020 LC-MS (Phenomenex Kinetex, 1.7 μm, 2.0 × 100 mm, C18 column) using positive and negative mode electrospray ionisation with a linear gradient of 5−95% MeCN - H$_2$O supplemented with 0.1% (v/v) formic acid in 15 min followed by 95% MeCN for 3 min with a flow rate of 0.3 mL/min. To convert desferriferricrocin into ferricrocin, FeCl$_3$ (final concentration at 1 mM) was added into the organic extract. To purify the fermentation product for structural analysis, similar extraction procedure was performed on 4 L cell

culture pellet. The organic extract was dried and dissolved in $H_2O$ and fractionated with Amberlite XAD-16 (Sigma-Aldrich) resin. The desferriferricrocin and ferricrocin were eluted from a gradient from 20% MeOH to 70% MeOH. The eluent was combined and purified by semipreparative HPLC using a reverse-phase column (Phenomenex Kinetics, C18, 5 μm, 100 Å, 250 × 4.6 mm). The identity of desferri-ferricrocin was confirmed by HR-MS and NMR analysis. The NMR spectra data are consistent with the literature data[51]. $^1$H-NMR (500 MHz, $CD_3OD$): δ 8.36 (s,), 4.46–4.30 (m, overlap, 3H), 4.24 (d, $J = 17.1$ Hz, 1H, Gly $C_{\alpha}H_2$), 4.08, (m, overlap, 1H, Ser $C_{\alpha}H$), 4.07 (d, $J = 16.0$ Hz, 1H, Gly $C_{\alpha}H_2$), 3.86 (ddd, $J = 56.5$, 11.1, 5.4 Hz, 2H Ser $C_{\beta}H_2$), 3.69 (d, $J = 15.7$ Hz, 1H, Gly $C_{\alpha}H_2$), 3.62 (d, $J = 17.0$ Hz, 1H, Gly $C_{\alpha}H_2$), 3.76-3.53 (m, overlap, 6H), 2.11 (s, 3H, hydroxamic $CH_3$), 2.105 (s, 3H, hydroxamic $CH_3$), 2.102 (s, 3H, hydroxamic $CH_3$), 2.00-1.90 (m, 1H, Orn $C_{\beta}H_2$), 1.90-1.85 (m, overlap, 1H, Orn $C_{\beta}H_2$), 1.85-1.80 (m, overlap, 1H, Orn $C_{\beta}H_2$),1.80-1.75 (m, overlap, 1H, Orn $C_{\beta}H_2$), 1.78-1.73 (m, overlap, 1H, Orn $C_{\beta}H_2$), 1.75-1.70 (m, overlap, 1H, Orn $C_{\beta}H_2$), 1.74-1.60 (m, overlap, 6H, Orn $C_{\beta}H_2$). $^{13}$C-NMR (125 MHz, $CD_3OD$): δ 174.6 C, 174.4, 174.2, 173.8 (overlap), 173.7, 172.5, 171.90, 171.88, 62.3 (Ser $C_{\beta}$), 56.7(Orn$_{\alpha}$), 56.2 (Ser $C_{\alpha}$), 54.9 (Orn $C_{\alpha}$), 54.5 (Orn $C_{\alpha}$), 48.41 (Orn $C_{\delta}$), 48.39 (Orn $C_{\delta}$), 48.3 (Orn $C_{\delta}$), 44.4 (Gly $C_{\alpha}$), 43.7(Gly $C_{\alpha}$), 30.3 (Orn $C_{\beta}$), 30.2 (Orn $C_{\beta}$), 28.3 (Orn $C_{\beta}$), 24.4 (Orn $C_{\gamma}$), 24.3(Orn $C_{\gamma}$), 24.0(Orn $C_{\gamma}$), 20.31 (hydroxamic $CH_3$), 20.29 (hydroxamic $CH_3$), 20.22(hydroxamic $CH_3$). HRMS: calc. for $[M + H]^+$ $C_{28}H_{48}N_9O_{13}^+$, 718.3367, found 718.3365.

### Synthesis of L-AHO amino acid substrate

The amino acid $N^5$-acetyl-$N^5$-hydroxy-L-ornithine (L-AHO) is synthesised from $N^2$-Cbz-$N^2$-Boc-L-ornithine according to the literature[35,36]. $^1$H-NMR (500 MHz, $CD_3OD$): δ 3.65 (t, $J = 6.0$ Hz, 1H, $C_{\alpha}H$), 3.56 (t, $J = 6.7$ Hz, 2H, $C_{\delta}H_2$, cis-trans isomers not resolved), 2.04 (s, 3H, acetyl $CH_3$, with a small shoulder at 2.01 due to cis-trans isomerization), 1.83-1.56 (m, overlap,.4H, $C_{\beta}H_2$, $C_{\gamma}H_2$). $^{13}$C-NMR (125 MHz, $CD_3OD$): δ 174.3 (carboxylate), 173.7 (amide carbonyl), 169.3 (amide carbonyl, minor isomer), 54.3 ($C_{\alpha}$), 50.9 ($C_{\delta}$, minor isomer), 47.2 ($C_{\delta}$, major isomer), 27.5 ($C_{\beta}$), 27.3 ($C_{\beta}$, minor isomer) 22.1 ($C_{\gamma}$), minor isomer), 19.7 (acetyl $CH_3$, minor isomer), 19.4 (acetyl $CH_3$). HRMS: calc. for $[M + H]^+$ $C_7H_{15}N_2O_4^+$, 191.1027; found 191.1097.

### Biochemical characterisation of SidC in vitro

Purified SidC and associated variants/mutants were converted to their holo- form by incubation in 20 mM Tris HCl, 100 mM NaCl, 2 μM of NpgA, 0.1 mM CoA and 10 mM $MgCl_2$ in a total volume of 50 μL for 1 h at 25 °C. Reactions were initiated by addition of ATP (5 mM) and either all or various combinations of the following: L-AHO (1 mM)/L-Ser (1 mM)/Gly (1 mM) in a final volume of 50 μL, and the reaction was allowed to proceed at 25 °C. At different time points, the reaction was quenched by mixing with an equal volume of methanol. The reaction products were analysed on an UHPLC-MS on a Shimadzu 2020 EVLC−MS controlled using Shimadzu LabSolutions (Phenomenex kinetex, 1.7 μm, 2.0 × 100 mm, $C_{18}$ column) using positive and negative mode electrospray S5 ionisation with a linear gradient of 5−95% MeCN −$H_2O$ supplemented with 0.1% (v/v) formic acid in 15 min followed by 95% MeCN for 5 min with a flow rate of 0.3 mL/min. All data were analysed using Shimadzu LabSolutions.

### Biochemical assays to determine adenylation domain specificity
**ATP-PPi exchange assays.** The amino acid substrate specificity profiles for the SidC $A_1$ and SidC $C_3A_3$ constructs were conducted using ATP-PPi exchange assays. Assays performed in 100 μL of reaction buffer (50 mM Tris-HCl pH 8, 2 mM $MgCl_2$) containing 1 mM TCEP, 5 mM ATP, 1 mM tetrasodium pyrophosphate ($Na_4PPi$), 5 mM substrate, and 5 μM enzyme. Before the addition of enzyme, $Na_4[^{32}P]$-PPi was added to a final intensity of ∼$2.5 \times 10^6$ cpm/mL. Reactions were allowed to proceed for 2 h at 25 °C and then quenched by the addition of 500 μL of charcoal

(3.6% w/v activated charcoal, 150 mM $Na_4PPi$, 5% $HClO_4$). Samples were centrifuged, and supernatant was discarded. To remove residual free $[^{32}P]PPi$, the pellet was washed twice with wash solution (0.1 M $Na_4PPi$, 5% $HClO4$). The pellet was resuspended in 500 μL of water and added to scintillation fluid at a final volume of 5 mL. Radioactivity was measured using a Beckman LS 6500 scintillation counter.

**Hydroxylamine Release Assays.** The hydroxylamine-trapping assay for detecting adenylation activity was conducted for the SidC $C_2A_2$ construct, and performed according to a reported protocol[38]. Briefly, the reaction was initiated by mixing 150 μL of substrate mixture [50 mM Tris, (pH 8.0), 30 mM $MgCl_2$, 300 mM hydroxylamine (pH 8.0), 10 mM carboxylic acid substrate] with an equal volume of enzyme mixture [100 mM Tris (pH 8.0), 20 mM ATP, 20 μM enzyme]. For some hydrophobic substrates, 2−5% (v/v) DMSO was included to facilitate dissolving the substrate. The reaction mixture was then incubated at 30 o C for 16 h. The reaction was stopped by mixing with 300 μL of stopping solution [10% (w/v) $FeCl_3 \cdot 6H_2O$ and S5 3.3% TCA dissolved in 0.7 M HCl]. The precipitated enzymes were removed by centrifugation at 17,000$g$ for 5 min, and 200 μL of the supernatant were transferred to a 96-well plate and the absorbance of the ferric-hydroxamate complex at 540 nm was measured by using a Tecan M200 plate reader.

### Biochemical characterisation of intra-molecular L-AHO loading by SidC $A_3$ domain
Purified SidC $C_3A_3T_3/C_3A_3T_3C_4T_4$ and $C_3A_3T_3^0C_4T_4$ proteins were converted to their holo- form by incubation in 20 mM Tris, 100 mM NaCl, 2 μM Sfp PPtase, 1 mM CoA and 10 mM $MgCl_2$ in a total volume of 50 μL for 1 h at 25 °C. Loading of L-AHO was initiated by addition of ATP (5 mM) and L-AHO (1 mM) in a final volume of 50 μL, and the loading reaction was allowed to proceed for 1 h at 25 °C before intact protein analysis by UHPLC-ESI-Q-TOF-MS.

### Biochemical characterisation of inter-molecular L-AHO loading by SidC $A_3$ domain
Purified SidC $C_3A_3T_3/C_4T_4/T_4/C_5T_5C_T$ and $T_5C_T$ proteins were converted to their holo- form by incubation in 20 mM Tris, 100 mM NaCl, 2 μM Sfp PPtase, 1 mM CoA and 10 mM $MgCl_2$ in a total volume of 50 μL for 1 h at 25 °C. Loading of L-AHO was initiated by the addition of ATP (5 mM) and L-AHO (1 mM) to a solution of holo-$C_3A_3T_3$ (100 μM) and one of holo-$C_4T_4/T_4/C_5T_5C_T$ and $T_5C_T$ (100 μM). The loading reaction was allowed to proceed for 1 h at 25 °C before intact protein analysis by UHPLC-ESI-Q-TOF-MS.

### Biochemical characterisation of intra-molecular Gly loading by SidC $C_2A_2T_2$
Purified SidC $C_2A_2T_2$ was converted to its holo- form by incubation in 20 mM Tris, 100 mM NaCl, 2 μM Sfp PPtase, 1 mM CoA and 10 mM $MgCl_2$ in a total volume of 50 μL for 1 h at 25 °C. To a solution of SidC $C_2A_2T_2$ (100 μM), loading of Gly was initiated by the addition of ATP (5 mM, or limited to a 2:1, 4:1 ratio with protein) and Gly (1 mM, or limited to a 2:1, 4:1 ratio with protein) in a total volume of 50 μL at 25 °C. Loading reactions were allowed to proceed for various time intervals before intact protein analysis by UHPLC-ESI-Q-TOF-MS.

### Biochemical characterisation of inter-molecular condensation reaction between SidC $C_2A_2T_2$-$Gly_3$ and SidC $C_3A_3T_3$-L-AHO
Purified SidC $C_2A_2T_2$ and SidC $C_3A_3T_3$ proteins were converted to their holo- form by incubation in 20 mM Tris, 100 mM NaCl, 2 μM Sfp PPtase, 1 mM CoA and 10 mM $MgCl_2$ in a total volume of 50 μL for 1 h at 25 °C. Reactions were initiated by addition of ATP (5 mM), L-AHO (1 mM), Gly (1 mM) to a solution containing holo-$C_2A_2T_2$ (50 μM) and holo-$C_3A_3T_3$ (100 μM) in a total volume of 50 μL at 25 °C. Reactions were allowed to proceed for either 10 min or 60 min before intact protein analysis by UHPLC-ESI-Q-TOF-MS. A variation of this reaction was conducted

which allowed $Gly_3$ and $Gly_5$ to be formed in situ on SidC $C_2A_2T_2$ (following procedure for *intra*-molecular Gly loading by SidC $C_2A_2T_2$) before its addition (at a final concentration of 50 μM) to a solution containing *holo*-$C_3A_3T_3$ (100 μM), ATP (5 mM), L-AHO (1 mM).

## UHPLC-ESI-Q-TOF-MS analysis of intact proteins

Biochemical assays were analysed on a Bruker MaXis II ESI-Q-TOF-MS connected to a Dionex 3000 RS UHPLC fitted with an ACE $C_4$–300 RP column (100 × 2.1 mm, 5 μm, 30 °C), controlled using Bruker Otof control 4.0. The column was eluted with a linear gradient of 5–100% MeCN containing 0.1% formic acid over 30 min. The mass spectrometer was operated in positive ion mode with a scan range of 200–3000 *m/z*. Source conditions were: end plate offset at −500 V; capillary at −4500 V; nebuliser gas ($N_2$) at 1.8 bar; dry gas ($N_2$) at 9.0 L min⁻¹; dry temperature at 200 °C. Ion transfer conditions were: ion funnel RF at 400 Vpp; multiple RF at 200 Vpp; quadrupole low mass at 200 *m/z*; collision RF at 2000 Vpp; transfer time at 110.0 μs; pre-pulse storage time at 10.0 μs. All spectra were analysed using Bruker DataAnalysis 4.4. Measured masses for all species are displayed in Table S2 and S3.

## AlphaFold modelling of SidC fragments

Structural models of the $A_3T_3C_4T_4$ and $C_4T_4C_5T_5$ regions of SidC were constructed using AlphaFold[39]. The full amino acid sequences of the excised tri-/tetra-domain regions were submitted to the AlphaFold Colab notebook (v1.5.2)[52], which uses a slightly simplified version of AlphaFold v2.3.1, with the run_relax parameter enabled. Structures were assessed for their reliability via inspection of PAE and pLDDT plots. The resulting structures were then aligned to each other via the $C_4$ domain present in both structures using PyMOL v1.3, yielding the final model of SidC $A_3T_3C_4T_4C_5T_5$. Structure co-ordinate files for the SidC $A_3T_3C_4T_4C_5T_5$ region and associated fragments used to assemble the model are available for download from Mendeley Data https://doi.org/10.17632/c3ymyp3yx4.1.

## Reporting summary

Further information on research design is available in the Nature Portfolio Reporting Summary linked to this article.

## Data availability

The minimum dataset necessary to interpret, verify and extend the work is provided in the manuscript and supplementary information. The raw data for Figs. 2 and 3, and Supplementary Figs. 11, 12 and 16, which were processed via standard deconvolution, are available upon written request to the corresponding authors. GenBank accessions have been provided for SidC (XM_653119, [https://www.ncbi.nlm.nih.gov/nuccore/XM_658335.1]), SidA (XM_658335, [https://www.ncbi.nlm.nih.gov/nuccore/XM_658335.1] and SidL (XM_652967, [https://www.ncbi.nlm.nih.gov/nuccore/XM_652967]) in the methods section. The DNA and amino acid sequence of the full-length SidC construct used in this study is reported in the supplementary information. Co-ordinate files for structural models of SidC have been deposited in Mendeley Data https://doi.org/10.17632/c3ymyp3yx4.1. A reporting summary for this Article is available as a supplementary information file.

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

## Acknowledgements

This work was supported by an NIH 1R35GM118056 grant to Y.T. Y.H. was supported by a Life Sciences Research Foundation Fellowship sponsored by the Mark Foundation for Cancer Research, and also a general start-up fund supported by UCSB. M.J. is the recipient of a BBSRC Discovery Fellowship (BB/R012121/1) and a UKRI Future Leaders Fellowship (MR/W011247/1), and M.P. is supported by a Midlands Integrative Bioscience Doctoral Training Partnership studentship (BB/M01116X/1). The Bruker MaXis II instrument used in this study was funded by the BBSRC (BB/M017982/1). R.R.O.L. and H.H.N. acknowledge funding from the U.S. Department of Energy (DOE) Office of Science (BER) contract DE-FC-02-661 02ER63421 (PI Yeates; UCLA/DOE Institute for Genomics and Proteomics). W.Z. and W.S. acknowledge funding from NIH DP2AT009148 grant. N.K.G. acknowledges shared instrumentation grants from the NSF (CHE-1048804) and the National Center for Research Resources (S10RR025631).

## Author contributions

Y.H., M.J. and Y.T. designed the study. Y.H. generated expression constructs and performed heterologous production of siderophores. Y.H., M.J. and M.P. overexpressed and purified proteins, constructed

mutant plasmids and performed biochemical assays. M.J., R.R.O.L. and H.H.N. conducted mass spectrometry analysis of intact proteins. J.K. and N.K.G. synthesised L-AHO. W.S. and W.Z. conducted adenylation domain activity assays. M.P. and M.J. constructed and analysed AlphaFold models. M.J. and Y.H. wrote the manuscript with input from all authors.

## Competing interests

The authors declare no competing interests.
