## [Peer Review File · Nature Communications]

REVIEWER COMMENTS

Reviewer #1 (Remarks to the Author):

The manuscript "Elucidating the molecular programming of a nonlinear nonribosomal peptide synthetase responsible for fungal siderophore biosynthesis" is recommended for publication with revisions as follows:

- Figure 2: The first marked peaks on Figure 2.b mass spectra are barely above the noise level with no clear apex. Same for the second mass in Figure 2.c bottom spectrum. How did the author ascertain these are real masses and not due to deconvolution artifacts or simply noise?

- Table S2: large mass errors for some of the identified species are not acceptable. For example, 117,821 Da vs 117,832 Da for SidC C5T5CT holo-.

Reviewer #2 (Remarks to the Author):

The study by Jenner et al elucidated the molecular programming of a nonlinear nonribosomal peptide synthetase (NRPS) involved in biosynthesis of the siderophore ferricrocin by the mold *Aspergillus nidulans*. Nonribosomal peptides play important roles as drugs, in microbial interaction and virulence. However, the molecular programming of nonlinear NRPS is poorly understood. Partly based on previous work, in which the same authors clarified how another fungal NRPS, SidD, assembles the siderophore fusarinine C in an iterative manner, this study provides novel and valid insights into the biochemistry of nonlinear NRPS. I have only minor critiques:

Page 4, line 25: please explain the abbreviation L-haOrn

Page 4, line 26: please explain the meaning of the -npgA suffix

Page 4, lines 2-3: this is not true for the NRPS Sid1 that is involved in synthesis of AS2488059 (Supplementary Figure S16. This is a particularly interesting NRPS as it incorporates quite unusual amino acids apart from acetyl-hydroxyornithine. I assume that incorporation of the required amino acids requires conservation of at least four A domains in contrast to other NRPS. This could be

discussed in more detail. Is it possible to predict or exclude substrate specificities of the A domains of this NRPS in the light of the new findings. Supplementary Figure shows an E4 domain: please explain.

Page 5, line 7: "synthesized according to literature protocols": please include References

Page 5, line 22: There are certain Ferrichrom-type siderophores that contain

Figure 2 legends states: "Presented with either a cellular pool of amino acids,..." This means the in vivo experiment, right? Please clarify.

Reviewer #3 (Remarks to the Author):

Comments:

In the manuscript "Elucidating the molecular programming of a nonlinear nonribosomal peptide synthetase responsible for fungal siderophore biosynthesis", Jenner, Hai, and colleagues described the biochemical characterization of the SidC NRPS, responsible for construction of the intracellular siderophore ferrichrome. First, the authors performed in vitro reconstruction of purified SidC. Coupled with intact protein mass spectrometry, the authors discovered several non-canonical processes during peptidyl siderophore biosynthesis, including inter-modular loading of amino acids and an adenylation domain capable of poly-amide bond formation. The authors claim that this work expands the scope of NRPS programming, allows biosynthetic assignment of ferrichrome NRPSs, and sets the stage for reprogramming towards novel hydroxamate scaffolds.

The manuscript is well written and claims are well supported by the presented data. The SI is fine. This reviewer understands that the data uncover several non-canonical events during peptidyl siderophore biosynthesis using conventional intact protein mass spectrometry. However, this reviewer struggles with evaluating the novelty presented in this manuscript. The novelty in this manuscript is limited to the biochemical characterization of the SidC NRPS, expanding the repertoire of fungal NRPS enzymology. The reviewer believes that structural data and/or engineering efforts would strengthen the manuscript. The novelty presented in the manuscript in its current format, however, appears to be rather narrow for a journal like Nature Communications. More specific journals would be better suited.

Response to Reviewers' Comments - NCOMMS-22-42324-T

Our responses (in blue) to the reviewers' comments (in black) are as follows:

Reviewer 1:

1). Figure 2: The first marked peaks on Figure 2.b mass spectra are barely above the noise level with no clear apex. Same for the second mass in Figure 2.c bottom spectrum. How did the author ascertain these are real masses and not due to deconvolution artifacts or simply noise?

Some of the markers in Fig. 2b and 2c are simply to highlight where a given species would give a peak, if it were present, based on calculated masses or previously measured spectra. This is particularly the case for the T₅C_T spectrum (Fig. 2c, bottom), for which we have highlighted where the L-AHO-T₅C_T species would appear in the spectrum if it were present (it's not because transfer of L-AHO requires the N-terminal C domain). This is simply to help the reader see when the species is present, and when it is not.

In the case of Fig. 2b – we know the mass of the holo- species from previously measured spectra (see spectra included below). The 1 x L-AHO marker in Fig. 2b (top) is simply to highlight where this species would appear if it were present (we agree, the peak is just above the noise – but gives a mass congruent with 1 x L-AHO). We have added a statement in the legend of Fig. 2 to clarify these points.

Figure showing measured masses of apo- and holo-C₃A₃T₃C₄T₄ – allowing assignment of these peaks in subsequent spectra.

2). Table S2: large mass errors for some of the identified species are not acceptable. For example, 117,821 Da vs 117,832 Da for SidC C₅T₅C_T holo-.

Agreed – we thank the reviewer for their keen eye here. We have gone back to data relating to SidC C₅T₅C_T and found that the internal calibration had not been applied in these cases – hence the larger mass errors for the apo-, holo- and L-AHO species. This has been amended in Table S2, and the panel for Fig. 2c (top) has been replaced.

Reviewer 2:

1). Page 4, line 25: please explain the abbreviation L-haOrn

L-haOrn is an abbreviation for N^δ-acetyl-N^δ-hydroxy-L-ornithine, also known as L-AHO. We used L-AHO in the Introduction and then switched to L-haOrn in the Results / Discussion section. This was unintentional and is

confusing for the reader. We have therefore changed everything to L-AHO for consistency (and is the more commonly used abbreviation in the literature). We thank the reviewer for spotting this error.

2). Page 4, line 26: please explain the meaning of the -npgA suffix

The *-npgA* suffix refers to the Ppant transferase from *Aspergillus nidulans*, which is integrated into the yeast chromosome to ensure efficient phosphopantetheinylation of the proteins. We have added an explanatory statement in the manuscript.

3). Page 4, lines 2-3: this is not true for the NRPS Sid1 that is involved in synthesis of AS2488059 (Supplementary Figure S16). This is a particularly interesting NRPS as it incorporates quite unusual amino acids apart from acetyl-hydroxyornithine. I assume that incorporation of the required amino acids requires conservation of at least four A domains in contrast to other NRPS. This could be discussed in more detail. Is it possible to predict or exclude substrate specificities of the A domains of this NRPS in the light of the new findings. Supplementary Figure shows an E4 domain: please explain.

Agreed. We have added some additional analysis on this pathway in the Discussion section of the manuscript. We have elected not to go into detail on this 'exception' in the Introduction section in order to keep the narrative clear for the reader. We have also added a reference which reports the discovery of the AS2488059 NRPS (*ACS Chem. Biol.* **2022**, 17, 207–216) - its omission from the original submission was an oversight on our part (now REF [46]). It is worth noting that phylogenetic analyses conducted in this paper suggest a different evolutionary origin for the Sid1 NRPS from that of ferrichrome family NRPSs – we have therefore been careful not to speculate much further on this matter.

Yes – assuming a linear biosynthetic model, it is possible to assign the amino acid specificity for each A domain in the Sid1 NRPS based on our findings. We have added these to Fig. S16.

The E₄ domain is an epimerization domain, a fairly common feature of NRPS assembly lines. It epimerizes the L-Phe residue loaded onto the T₃ domain, giving rise to D-Phe in the final AS2488059 product. We have added an explanatory statement in the legend of Fig. S16.

4). Page 5, line 7: "synthesized according to literature protocols": please include References

These references have been added – [35] and [36].

5). Page 5, line 22: There are certain Ferrichrom-type siderophores that contain

We are unsure what this comment from the reviewer is referring to here. There is no passage in the manuscript that matches this phrase. Is the comment incomplete?

6). Figure 2 legends states: "Presented with either a cellular pool of amino acids,...." This means the in vivo experiment, right? Please clarify.

Correct – we have clarified in the figure legend.

Reviewer 3:

1). '...this reviewer struggles with evaluating the novelty presented in this manuscript. The novelty in this manuscript is limited to the biochemical characterization of the SidC NRPS, expanding the repertoire of fungal NRPS enzymology. The reviewer believes that structural data and/or engineering efforts would strengthen the manuscript.'

As the reviewer points out in their summary comments, this work reports the complete biochemical characterisation of a highly unusual siderophore-producing NRPS in fungi, which has been a long-standing

biosynthetic conundrum in the field. In doing so we have also discovered novel features of fungal NRPS enzymology, namely: i). the ability of an A domain to load multiple T domains up to n+2 modules downstream, and ii). an A domain that can catalyse amide bond formation with an α -NH₂ group. We would argue that both of these features are highly novel and set the stage for future engineering efforts to exploit these unique features of the NRPS.

It is unclear exactly what structural information/engineering experiments the reviewer is referring to here. We agree, structural data for the entire NRPS, or indeed individual domains (particularly the A₂ domain), would undoubtedly provide additional insights into how some of the novel features we have uncovered manifest themselves at the molecular level. However, obtaining structural data (X-ray or cryo-EM) is not trivial, and would require a dedicated campaign of work to achieve – we therefore believe this to be beyond the scope of our manuscript.

In the absence of experimental structural data, we have used AlphaFold to model fragments of the SidC NRPS to see if any insights could be obtained. Notably, modelling of the A₃T₃C₄T₄C₅T₅ region suggests that both C₄ and C₅ domains form interfaces with the A₃ domain, conceivably providing a platform for their respective T domains to access the A₃ domain active site, and supports the intra-chain model for A₃ loading of T domains. We have included this as an additional panel in Fig. S13 and added a section in the main text. Whilst it was possible to model the A₂ domain using AlphaFold and use this model to rationalize the unusual function of the A₂ domain, the conclusions we could draw would be incredibly speculative without concrete experimental data.

Regarding engineering efforts, our experiments in Fig. 3a use a truncated form of SidC, lacking the A₁T₁ domain, which gives rise to ferrichrome production rather than ferricrocin under the same conditions. We would argue that these represent initial engineering experiments on the SidC NRPS, which set the stage for a more dedicated effort to produce novel derivatives in the future.

Additions / Edits Outside of Reviewers Comments:

- We have added a Data Availability statement at the end of the manuscript.
- Coprogen is now mentioned in the Introduction as another example of an extracellular siderophore, with two accompanying references – [10] and [11].
- Correction to funding codes for MJ in the Acknowledgements section.

REVIEWER COMMENTS

Reviewer #1 (Remarks to the Author):

The authors adequately addressed my previous comments on the intact MS analysis and figures in the revision.

Reviewer #2 (Remarks to the Author):

All my comments and suggestions have been addressed to my satisfaction. Congratulation to this interesting study results!

Reviewer #3 (Remarks to the Author):

The authors have suitably revised their manuscript, taking up recommendations from both reviewers. The manuscript is ready for publication and I recommend its acceptance.

Reviewer #4 (Remarks to the Author):

As asked by the editor, I have only focused on the AlphaFold calculations.

AlphaFold was used through the ColabFold notebook, which uses a faster but only marginally less performant sequence search. A citation to ColabFold is missing (Mirdita et al, Nature Methods 2022;19:679-682). They should also specify which version of ColabFold was used.

The calculation was performed correctly, however, no information on plddt (predicted IDDT) is supplied, making it impossible to judge the reliability of the results. Plots of the PAE (predicted alignment error) are even better as they provide information as to the reliability of the relative domain organization. Such plots were already provided by the earliest versions of ColabFold.

Without this information, the model is worthless without additional experimental evidence and the statements in the main text too speculative.

A revised version would have to show plddt and pae plots, which can be in the same supplementary figure. In addition, the model itself (PDB coordinates) would be most useful for anybody wanting to explore this further.

Response to Reviewers' Comments - NCOMMS-22-42324-T

Our responses (in blue) to the reviewers' comments (in black) are as follows:

Reviewer 4:

As asked by the editor, I have only focused on the AlphaFold calculations.

1). AlphaFold was used through the ColabFold notebook, which uses a faster but only marginally less performant sequence search. A citation to ColabFold is missing (Mirdita et al, Nature Methods 2022;19:679-682). They should also specify which version of ColabFold was used.

We have added the ColabFold reference to the Materials and Methods section of the SI, and noted the version of ColabFold that was used (v1.5.2).

2). The calculation was performed correctly, however, no information on plddt (predicted IDDT) is supplied, making it impossible to judge the reliability of the results. Plots of the PAE (predicted alignment error) are even better as they provide information as to the reliability of the relative domain organization. Such plots were already provided by the earliest versions of ColabFold. Without this information, the model is worthless without additional experimental evidence and the statements in the main text too speculative.

A revised version would have to show plddt and pae plots, which can be in the same supplementary figure. In addition, the model itself (PDB coordinates) would be most useful for anybody wanting to explore this further.

Agreed. We have added PAE, pLDDT and Sequence Coverage plots for each of the SidC fragments modelled (**Fig. S14 and S15**). Furthermore, we have added a structure of the 'rank_1' model used for each fragment coloured according to the pLDDT score. Structural predictions for the core domains are 'very high (>90)' with small regions of 'confident (80)'. The short sections with a 'very low (<50)' score correspond to disordered linker regions between core domains, which are a common feature of NRPSs and to be expected (*Bioinformatics*. **2019**, 35, 3584–3591).

We have uploaded the .pdb co-ordinates for each fragment of SidC and the combined A₃T₃C₄T₄C₅T₅ model to Mendeley Data under the following DOI: 10.17632/c3ymyp3yx4.1.

We would like to thank the reviewer for the points raised here, they been very useful and set an excellent standard for reporting our use of AlphaFold in the future.

REVIEWERS' COMMENTS

Reviewer #4 (Remarks to the Author):

I am satisfied with the changes.

In addition, I would like to congratulate the authors on their nice study.